# Endoscopic liquid biopsies of gastric fluid in a large human patient cohort reveal DNA content as a candidate tumor biomarker in gastric cancer

Francine C Cadoná[1,2], Thais F Bartelli[1†], Adriane G Pelosof[3], Claudia Z Sztokfisz[3], Adriana P Bueno[4], Luana Batista do Carmo dos Santos[1], Gabriela P Branco[1], Gabriel Oliveira dos Santos[4], Warley A Nunes[4], Fernanda A Pintor[5], Laís Lie Senda de Abrantes[5], Alexandre Defelicibus[6], Luiz Gonzaga Vaz Coelho[7], Marcis Leja[8], Haejin In[9,10], Sharon Li[11,12], Howard Hochster[9,13], Felipe JF Coimbra[14], Rodrigo Drummond[6], Israel Tojal Da Silva[6], Ravi J Chokshi[11,15], Renata Pasqualini[11,16], Wadih Arap[11,17]*, Diana N Nunes[1,11,16], Emmanuel Dias-Neto[1,11,16]*

[1]Laboratory of Medical Genomics, A. C. Camargo Cancer Center, São Paulo, Brazil; [2]Masters in Health and Life Sciences, Franciscana University, Santa Maria, Brazil; [3]Endoscopy Section, A. C. Camargo Cancer Center, São Paulo, Brazil; [4]Department of Pathology, A. C. Camargo Cancer Center, São Paulo, Brazil; [5]Research Support Center, A. C. Camargo Cancer Center, São Paulo, Brazil; [6]Laboratory of Computational Biology, A. C. Camargo Cancer Center, São Paulo, Brazil; [7]Postgraduation Program in Applied of Sciences of Adult Health & Alpha Gatroenterology Institute, Hospital das Clínicas, Faculty of Medicine, Federal University of Minas Gerais, Belo Horizonte, Brazil; [8]University of Latvia, Institute of Clinical and Preventive Medicine and Center for Gastric Diseases GASTRO, Riga, Latvia; [9]Rutgers Cancer Institute, New Brunswick, New Brunswick, United States; [10]Division of Surgical Oncology, Department of Surgery, Rutgers Robert Wood Johnson Medical School, New Brunswick, United States; [11]Rutgers Cancer Institute, Newark, United States; [12]Division of Hematology/Oncology, Department of Medicine, Rutgers New Jersey Medical School, Newark, United States; [13]Division of Medical Oncology, Department of Medicine, Rutgers Robert Wood Johnson Medical School, New Brunswick, United States; [14]Department of Abdominal Surgery, A. C. Camargo Cancer Center, São Paulo, Brazil; [15]Division of Surgical Oncology, Department of Surgery, Rutgers New Jersey Medical School, Newark, United States; [16]Division of Cancer Biology, Department of Radiation Oncology, Rutgers New Jersey Medical School, Newark, United States; [17]Division of Hematology/Oncology, Department of Medicine, Rutgers New Jersey Medical School, Newark, United States

*For correspondence: wa116@rutgers.edu (WA); eyd10@rutgers.edu (ED-N)

Present address: †Department of Clinical Cancer Prevention, The University of Texas M. D. Anderson Cancer Center, Houston, United States

## eLife Assessment

This **valuable** work substantially advances our understanding of prognostic value of total gfDNA in gastric cancer. The evidence supporting the conclusions is **solid**, supported by a large, well-classified patient cohort and controlled clinical variables. The work will be of broad interest to scientists and clinical pathologist working in the field of gastric cancer.

**Abstract** Gastric cancer remains a diagnostic and therapeutic challenge worldwide. Improved prognostic biomarkers could aid treatment planning across surgical, neoadjuvant, and adjuvant settings. We evaluated a novel liquid-biopsy approach integrated with esophagogastroduodenoscopy (EGD) by analyzing gastric fluid DNA (gfDNA) from a large cohort (n=1056) to assess its diagnostic utility and prognostic value in gastric cancer. In this exploratory study, gfDNA concentration was measured in patients with normal gastric mucosa, peptic diseases, preneoplastic conditions, or cancer. Variables included sex, gastric fluid pH, proton-pump inhibitor use, tumor subtype, stage, and outcomes. gfDNA levels were significantly higher in gastric cancer than in all comparison groups (mean 26.86 ng/µL; 95% CI 20.05–33.79; p=$3.61 \times 10e^{-12}$) and as compared to non-malignant controls (mean 10.77 ng/µL; 95% CI 9.23–12.33; p=$9.55 \times 10e^{-13}$) and preneoplastic states (mean 10.10 ng/µL; 95% CI 7.59–12.60; p=$1.10 \times 10e^{-5}$). Advanced tumors (T3) exhibited higher gfDNA than earlier stages (T2 or below; mean 25.66 vs 15.12 ng/µL; p=$5.97 \times 10e^{-4}$). In a subset of gastric cancer patients, gfDNA >1.28 ng/µL associated with longer progression-free survival (p=0.009) and correlated with increased tumor-infiltrating immune cells (p=0.001); this association remained after adjusting for stage (p=0.014). Elevated gfDNA supports gastric cancer presence in the general human population and may inform disease management when combined with tissue biopsies. Importantly, gfDNA shows prognostic potential in established gastric cancer, where higher gfDNA content may paradoxically relate to better outcomes, potentially linked to immune-cell infiltration. These findings warrant further validation and integration with complementary diagnostic modalities to enhance accuracy and clinical utility.

## Introduction

Esophagogastroduodenoscopy (EGD) is routinely performed to work up vague digestive complaints and is used to biopsy suspicious lesions and ultimately diagnose gastric cancer, a tumor that ranks as the fifth in terms of incidence and mortality in the world, with current estimates of ~660,000 annual deaths globally (*Globocan/IARC, 2022*). In areas of high incidence, EGD has been shown to contribute to a reduction in gastric cancer mortality (*Hamashima et al., 2013*; *Matsumoto and Yoshida, 2014*) and is also employed for presurgical gastric cancer staging and even as a treatment method (endoscopic resection) in early-stage disease (*Kim and Jung, 2021*), rendering it a universally used tool in this setting. During EGD, gastric fluids are collected and discarded, allowing a better evaluation of the gastric mucosa.

The study of DNA found in body-fluid samples (e.g. blood, urine, saliva, cerebrospinal fluid, peritoneal washes) can support the diagnosis and/or follow-up of cancer (*Forshew et al., 2012*). In a liquid biopsy approach, tumor-derived DNA permits the monitoring of specific mutations diluted in the fluids under investigation. As the gastric fluid is in direct contact with the stomach mucosa and gastric cancer lesions, liquid biopsies performed with gastric fluid are neither restricted to just the few selected areas sampled by tissue biopsy, nor by the minute amounts of tumor DNA that might be diluted in peripheral blood. Also, it may indeed allow a full representation of cells that reach the stomach, derived from the upper digestive tract, including immune cells, tumor cells, and the microbiota. In a prior feasibility study, we showed that gastric fluid incidentally collected from gastric cancer patients during EGD contains tumor-derived DNA, allowing the capture of tumor mutations (*Pizzi et al., 2019*). We have now collected gastric fluids from a large patient cohort (n=1056), obtained during diagnostic endoscopy, to investigate whether gastric fluid DNA (gfDNA) concentrations would vary according to disease diagnosis, including the presence of non-malignant, premalignant, or malignant lesions, gastric cancer subtype and stage, tumor location, and patient age or sex. Notably, we also evaluated whether gfDNA concentrations would have predictive and prognostic value on gastric cancer recurrence and patient outcomes.

## Results

### Patient groups and diagnoses

Gastric fluids were collected and gfDNA was extracted from a total of 1056 subjects. After excluding 115 cases with a diagnosis of hepatic portal hypertension, partial or total gastrectomy, esophagectomy,

**Table 1.** Clinical and demographic attributes of participating subjects (n=941).

| | | Non-gastric cancer* (n=705, 74.9%) | Gastric cancer (n=236, 25.1%) |
|---|---|---|---|
| Age | ≤45 | 130 (18.4%) | 34 (14.4%) |
| | 45–60 | 224 (31.8%) | 83 (35.2%) |
| | ≥60 | 343 (48.7%) | 119 (50.4%) |
| | Missing data | 8 (1.1%) | 0 |
| Sex | Male | 330 (46.8%) | 150 (63.6%) |
| | Female | 367 (52.1%) | 86 (36.4%) |
| | Missing data | 8 (1.1%) | 0 |
| †BMI | Underweight | 10 (1.4%) | 9 (3.8%) |
| | Normal weight | 206 (29.2%) | 91 (38.6%) |
| | Overweight | 310 (44%) | 84 (35.6%) |
| | Obese | 177 (25.1%) | 51 (21.6%) |
| | Missing data | 2 (0.3%) | 1 (0.4%) |

*Includes all patients diagnosed as presenting normal mucosa, with peptic diseases or pre-neoplastic lesions.
†Body-mass index (BMI): Underweight, BMI <18.5; Normal weight, BMI = 18.5–24.9; Overweight, BMI = 25–29.9; Obese, BMI ≥30.

or other non-gastric cancer malignancies (e.g. esophageal squamous cell carcinoma, duodenal adeno-carcinoma, Hodgkin's disease, well-differentiated neuroendocrine tumor, and metastatic breast cancer, among others), a total of 941 patients (89.1%) remained for gfDNA analysis. These patients were grouped according to the EGD findings as: normal EGD (n=10, 1.1%), non-malignant peptic diseases (n=596, 63.3%) including erosive gastritis, erosive and non-erosive esophagitis with or without hiatal hernia, infectious or inflammatory disorders, preneoplastic conditions (n=99, 10.5%) including atro-phic gastritis, gastric intestinal metaplasia, dysplasia, Barrett's esophagus, or gastric adenocarcinoma (n=236, 25.1%). The diagnostic group, age, sex, and body-mass index (BMI) of the participants are shown in *Table 1*.

## Gastric fluid DNA concentrations according to age, pH, and body-mass index

No significant variations in gfDNA concentration were found by age, pH of gastric fluid, or BMI when examined by group amongst gastric cancer patients or non-cancer controls. Regarding age, subjects were clustered in the three groups (<45, 45–60, and >60 years old), and no significant differences were found between subjects with (p=0.26) or without (p=0.14) gastric cancer. Similarly, no difference was found by pH analysis of gastric fluid (grouped as pH <7, pH = 7, or pH >7); we found p-values of 0.60 and 0.90 for gastric cancer and non-cancer controls, respectively. For BMI analysis, subjects were grouped as underweight (BMI <18.5), normal (BMI between 18.5–24.9), overweight (BMI between 25–29.9), or obese (BMI ≥30). Again, no differences in gfDNA concentrations were found by BMI in individuals with gastric cancer (p=0.59) or non-cancer controls (p=0.25).

## gfDNA concentrations and sex

Next, we evaluated whether gfDNA concentrations vary according to sex in patients with gastric cancer and subjects without cancer. For the non-cancer cohort, we observed elevated gfDNA concen-trations in females versus males (p=6.8e$^{-4}$), but no differences were seen between males and females in the cancer group (p=0.7; *Figure 1*). Therefore, when we compared gfDNA concentration between cancer versus non-cancer subjects, considering sex, we found striking differences for men and for women, respectively, p=7.98e$^{-12}$ and p=1.19e$^{-4}$ (*Figure 1*).

As some studies have suggested that the use of proton pump inhibitors (PPI) is more frequent in women (*Rückert-Eheberg et al., 2022*), we evaluated possible links between gfDNA concen-trations and the reported history of current, previous, or no PPI use. In general, we observed no

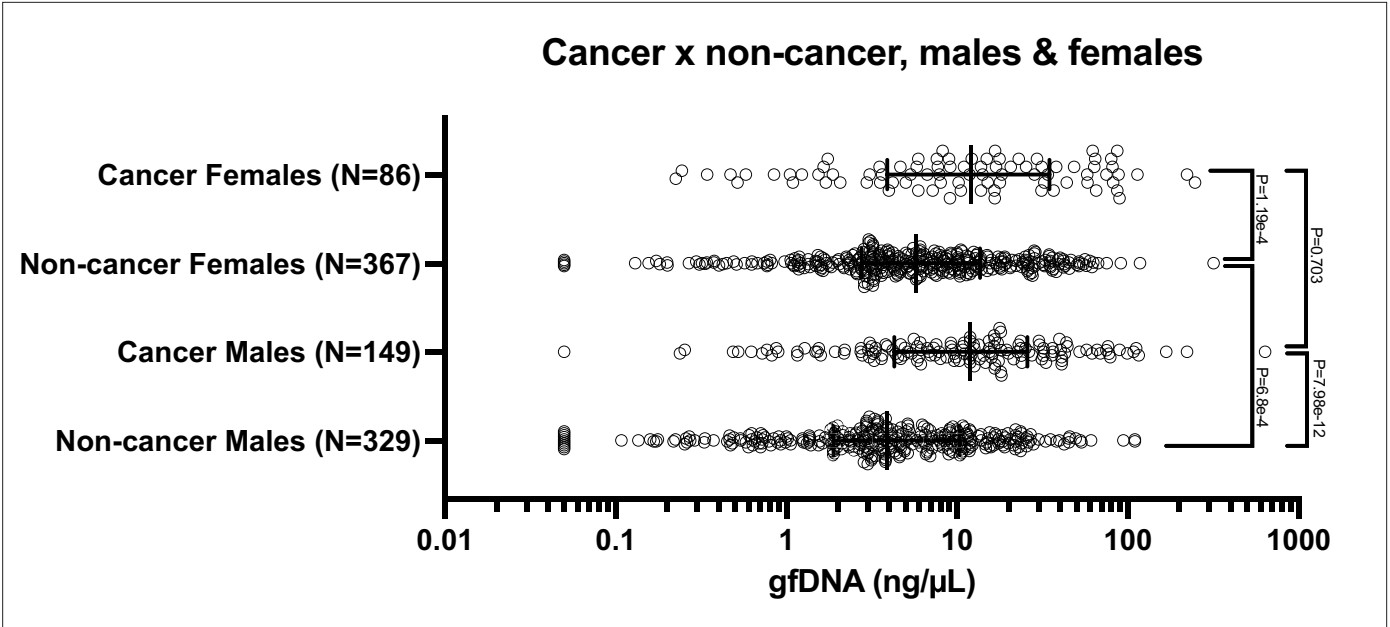

**Figure 1.** gfDNA concentrations observed in all subjects with available data, grouped according to sex, and presence of gastric cancer. Non-cancer subjects have normal gastric mucosa or minor peptic diseases (excludes pre-neoplastic disease). Statistical comparisons were performed using the Kruskal–Wallis test, followed by pairwise Mann–Whitney *U* tests with Benjamini–Hochberg correction. *P*-values < 0.05 were considered significant. Error bars indicate median gfDNA concentration (ng/μL) with interquartile ranges.

impact of PPI on gfDNA concentrations. However, for the group of non-cancer subjects that used PPI (n=410), increased gfDNA concentrations were seen for women (n=206), as compared to men (n=204; p=1.51e$^{-4}$), apparently reflecting the sex-related results shown in *Figure 1*, between males and females, unrelated to PPI use. Moreover, gfDNA concentrations were not different for subjects that reported current and previous administration, as well as no PPI administration, for females with or without cancer (n=55, n=6 and n=25; n=111, n=45 and n=206, respectively), as well as for males with or without cancer diagnosis (n=60, n=19 and n=62; n=75, n=51 and n=209, respectively, with corresponding p-values of p=0.92; p=0.07; p=0.24, and p=0.71).

## Analysis of bacteria to human ratios in gfDNA

As the gfDNA origin is likely to be a sum of human DNA (including epithelial, stromal, and immune cells, tumor and non-tumor cells), combined with microbiota-derived DNA (transient, saliva-derived microorganisms, the resident microbiota of the stomach and other upper digestive tract structures), we performed a quantitative analysis of bacteria and human DNA for a subset of samples (n=180), based on human-to-bacteria DNA ratios, as recently described (*de Albuquerque et al., 2022*). The samples selected for the analysis of this subset analysis included all 10 controls without any pathologies detected, as well as samples from the peptic diseases group (n=51), preneoplastic conditions (n=55), and the gastric cancer group (n=64). This analysis revealed no significant differences in the ratios of human- or bacteria-derived DNA found in the gfDNA (*Supplementary file 1*).

## gfDNA concentrations are increased in gastric cancer patients compared to non-cancer controls

We first determined the concentration of gfDNA from subjects with different diagnoses after EGD and found the following values for the groups: normal mucosa and peptic diseases - mean 10.77 ng/μL; 95% CI: 9.23–12.33; n=606; preneoplastic conditions - mean 10.10 ng/μL; 95% CI: 7.59–12.60; n=99 and gastric cancer - mean 26.86 ng/μL; 95% CI: 20.05–33.79; n=236. Notably, mean concentration of gfDNA was significantly different among the groups (p=3.61e$^{-12}$). gfDNA concentration was higher in gastric cancer as compared to the combined group of normal EGD and peptic disease patients (p=9.55e$^{-13}$), as well as to the preneoplastic group (p=1.10e$^{-5}$). In contrast, gfDNA concentrations were not different between the non-cancer groups (p=0.89; *Figure 2A*).

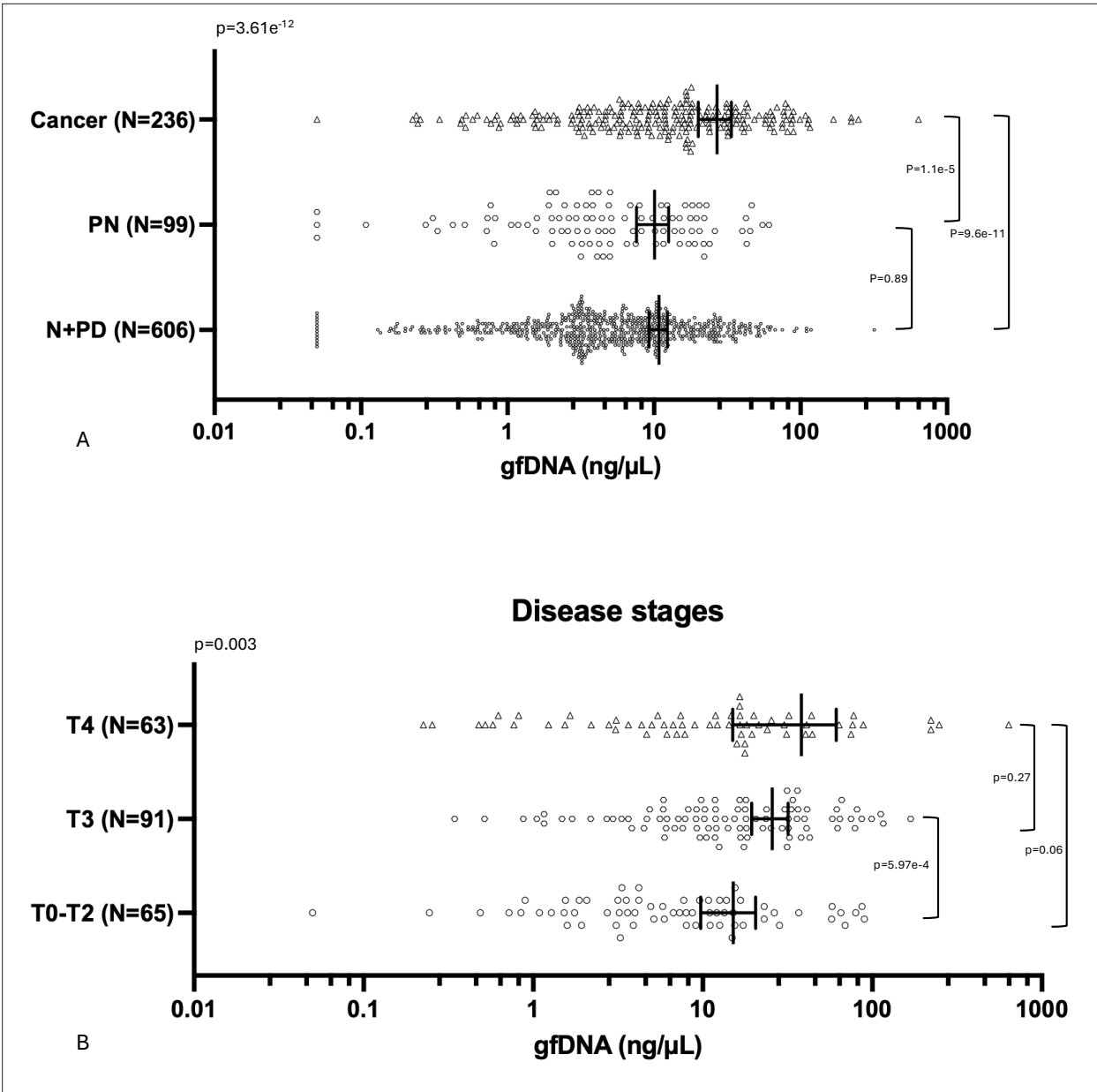

**Figure 2.** gfDNA concentration according to disease diagnosis and tumor stage. (**A**) gfDNA concentration (ng/μL) in patients with no endoscopic findings: Normal (N) or presenting minor peptic diseases (PD) – N+PD; preneoplastic conditions (PN), or gastric cancer (Cancer). (**B**) gfDNA concentrations (ng/μL) for early-stage disease patients – T0-T2 (T0 +Tis + T1+T2), as compared to more advanced disease stages T3 and T4. Statistical comparisons were performed using the Kruskal–Wallis test, followed by pairwise Mann–Whitney *U* tests with Benjamini–Hochberg correction. *P*-values < 0.05 were considered significant. Error bars indicate mean gfDNA concentration (ng/μL) and 95% confidence intervals.

For gastric cancer patients, we evaluated possible correlations between gfDNA concentrations and clinicopathological variables such as tumor stage and histopathological grades. In general, increments in gfDNA were found in subjects as the disease progresses (*Figure 2B*). Notably, the gfDNA mean concentration for early-stage disease (tumor stages - T0 +Tis + T1+T2, or T0-T2 n=65) was 15.12 ng/μL; 95% CI: 9.73–20.50, increasing in T3 (25.66 ng/μL; 95% CI: 19.46–31.85; n=91; p=5.97e$^{-4}$) and T4 stages (38.12 ng/μL; 95% CI: 15.02–61.22; n=63; p=0.06). Other comparisons were not significant, including gfDNA concentrations for patients with localized (n=165) versus metastatic disease (n=71). gfDNA concentrations could not differentiate tumor histopathological grades (p>0.05), or Lauren gastric cancer subtypes: diffuse (n=97), intestinal (n=93), or mixed (n=25; p=0.28; data not shown).

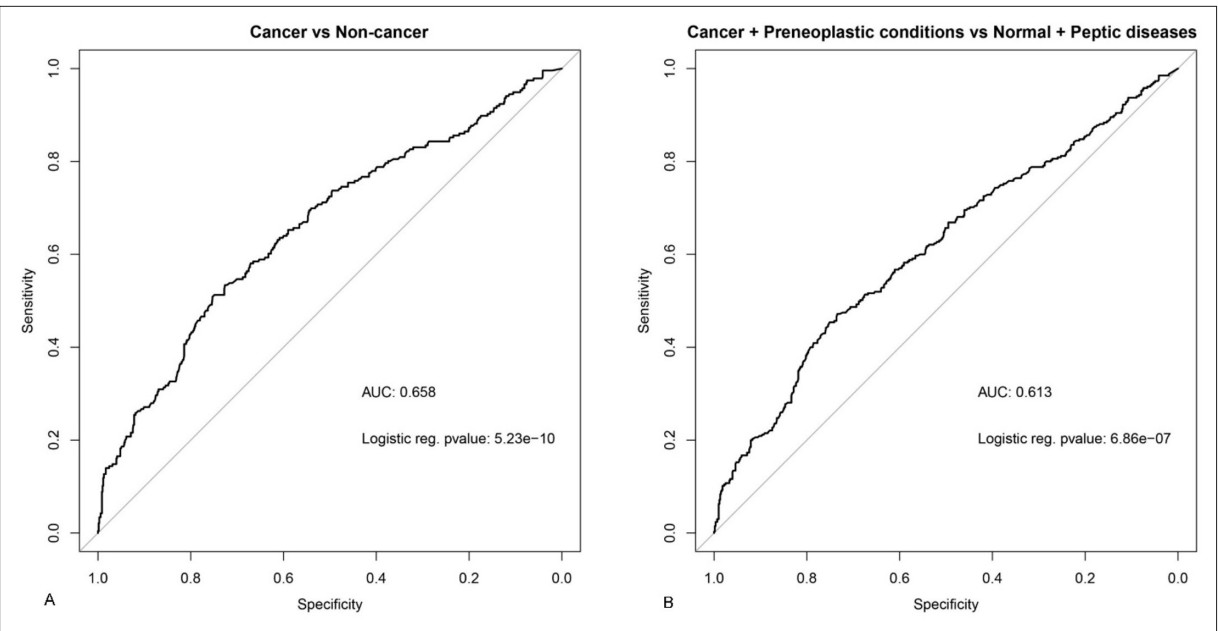

**Figure 3.** Analysis of receiver operating characteristics (ROC) and area under the curve (AUC) for gfDNA levels between cancer patients and non-cancer individuals. (**A**) ROC curve of gfDNA between the cancer patients and non-cancer individuals. (**B**) ROC curve of gfDNA between cancer versus non-cancer and cancer + preneoplastic conditions versus peptic diseases + normal.

## gfDNA correlates with the presence of inflammatory cell infiltrates and gastric cancer diagnosis and survival

Our liquid biopsy results revealed increased gfDNA concentrations in gastric cancer patients as compared to non-cancer controls. To explore whether gfDNA concentrations could be used to support gastric cancer diagnosis, we analyzed the receiver operating characteristic (ROC) curve of gfDNA between gastric cancer versus non-cancer controls. We observed a statistically significant curve (area under the curve [AUC] = 0.66; p=5.23e$^{-10}$), with a fair capability of detecting true positive gastric cancer patients by simply considering gfDNA concentrations (*Figure 3A*; *Supplementary file 2*), indicating its fair diagnostic support value. We also found a statistically significant curve of gfDNA between cancer/preneoplastic conditions and peptic diseases/normal (AUC = 0.61; p=6.86e$^{-7}$; *Figure 3B*).

We next evaluated if gfDNA concentrations would correlate with disease prognosis. Well-annotated clinical records were collected for all subjects to determine gfDNA concentrations cutoff values that would split the patients into groups with contrasting overall survival through a log-rank test. By using this approach, a gfDNA cutoff of 1.28 ng/μL was set and gastric cancer patients presenting gfDNA concentrations above this threshold had better overall survival (n=163; p=0.019), even more significant after patients with metastatic tumors were not considered (n=148; p=0.014; *Figure 4A*). Finally, to further investigate the possible link between patient survival and gfDNA concentration, we examined a subset of gastric cancer patients (n=32), for which pathology slides were available, to investigate possible correlations between gfDNA concentrations and the intensity of inflammatory cell infiltrates (*Figure 4B*). For this, we managed to include eight samples with gfDNA below the threshold and 24 above it. Even with a small sample set, this result was also significant (p=0.001; *Figure 4C* and *Supplementary file 3*), suggesting a possible mechanism to support the correlation of gfDNA concentration and gastric cancer survival data (*Figure 4A*).

## Discussion

Gastric cancer remains a leading cause of tumor mortality worldwide, a clinical feature attributable at least in part to its often-late stage at diagnosis and/or lack of prognostic factors to enable personalized disease management. However, with varying access to subspecialists, a role for gfDNA may

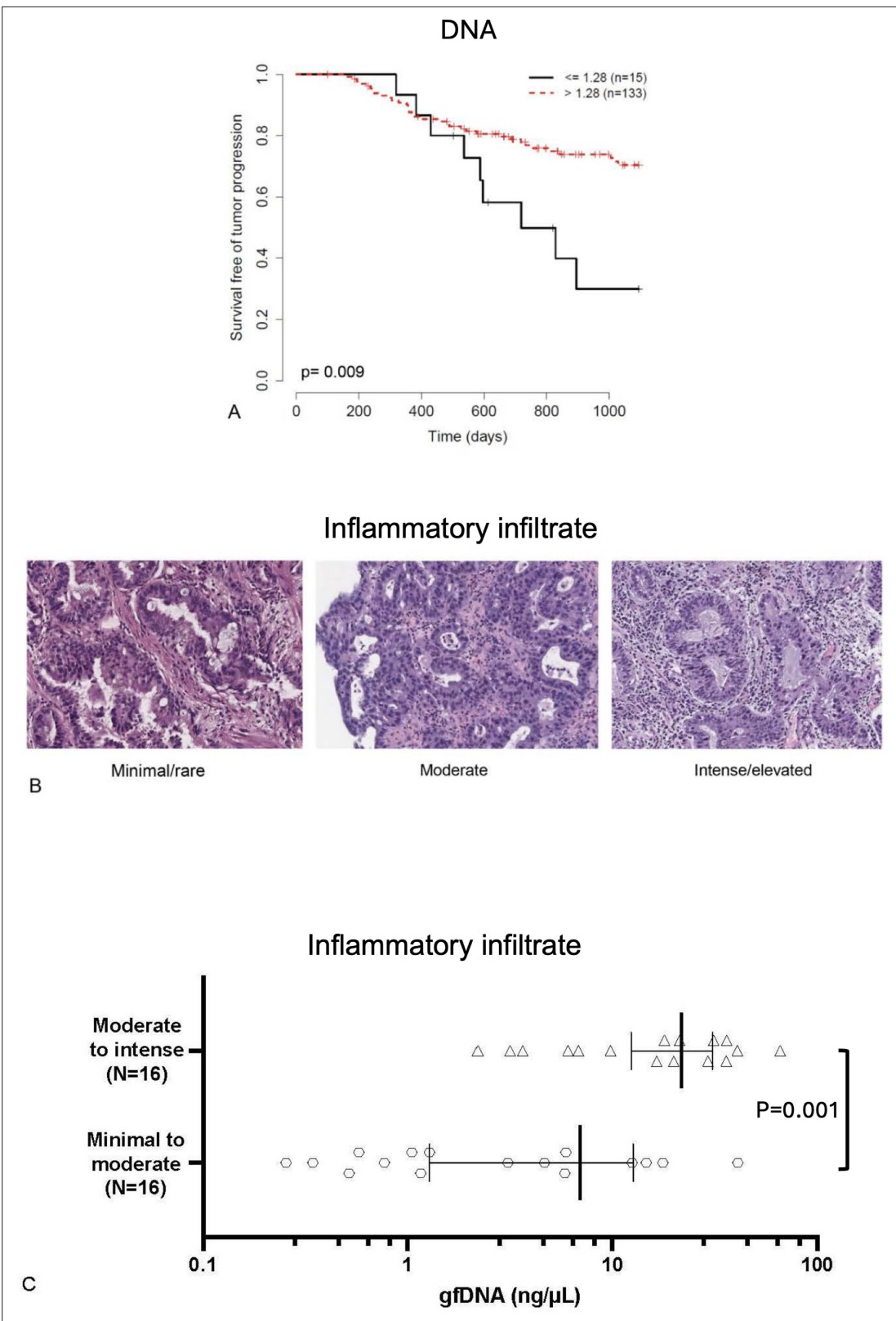

**Figure 4.** gfDNA analysis according to patient survival and infiltration of inflammatory cells. (**A**) For a total of 148 non-metastatic GC patients, cutoff gfDNA concentrations determined by a log-rank test (1.28 ng/μL) discriminate survival free of tumor progression (p=0.009). (**B**) For a subset of 32 cases, for which representative biopsy H&E slides were available, inflammatory cell infiltrates in the gastric tumors were inspected by experienced pathologists and (**C**) have shown that more intense inflammatory infiltrates in tumors are characteristic of subjects with higher gfDNA concentrations (p=0.001).

*Figure 4 continued on next page*

*Figure 4 continued*

gfDNA value distributions were compared using the non-parametric Mann–Whitney *U* test. *P*-values < 0.05 were considered statistically significant. Error bars indicate mean gfDNA concentration (ng/µL) and 95% confidence intervals.

help in the clinical management of patients. Early gastric cancer detection may often be delayed during EGD if biopsy samples are either taken too superficially or if the tissue is too scanty for meaningful pathological analysis. Thus, the translational application of a biomarker that provides support to gastric cancer diagnosis and prognosis would be useful, especially in a less specialized outpatient ambulatory or inpatient hospital settings.

In *Leon et al., 1977*, published a seminal paper showing that cancer patients had higher levels of cell-free DNA (cfDNA) in their blood compared to healthy individuals, suggesting its use as a potential cancer biomarker. Since then, numerous studies have confirmed these findings and have explored the potential of using cfDNA for cancer diagnosis, prognosis, and monitoring treatment response, including gastric cancer. *Qian et al., 2017* analyzed cfDNA in serum samples of subjects with gastric dysfunctions, including gastric cancer, and found that cancer patients presented elevated cfDNA when compared to benign gastric disease or controls. However, the diluted amounts of tumor-derived/tumor-microenvironment DNA remain minute in the peripheral circulation. This prompted us to determine DNA concentrations in the gastric fluids of subjects undergoing EGD to investigate non-specific digestive symptoms. Patients included those with diverse peptic diseases (controls), pre-neoplastic lesions, as well as patients with confirmed gastric cancer. We reasoned that liquid biopsies of gastric fluid incidentally obtained during EGD would offer diagnostic support and would possibly offer clinically useful information. Whereas EGD is an invasive procedure, most of these patients will receive at least one diagnostic EGD during their initial workup and tumor staging, and gastric fluids collected at this moment can be useful. However, prior studies of gastric fluids have heretofore been limited to the evaluation of certain molecules, such as proteins, RNAs, or for the analysis of mutations in specific genes (*Wu and Chung, 2013*; *Shao et al., 2014*; *Pizzi et al., 2019*). Here we show that the simple evaluation of gfDNA concentration serves as a convenient and promising biomarker to support gastric cancer diagnosis and staging and correlates with immune cell infiltrates. In a large cohort, we show that increased DNA concentrations in gastric fluids may differentiate gastric cancer patients from individuals without cancer, independent of age, BMI, sex, PPI use, or Lauren's histological subtypes. Increasing amounts of gfDNA were significantly associated with gastric cancer diagnosis, as well as advanced tumor stages.

Of note, there were several interesting observations from our results. Overall, we found that T4 tumors had gfDNA concentration similar to T3 tumors (T3: 25.66 ng/µL; 95% CI: 19.46–31.85; and T4: 38.12 ng/µL; 95% CI: 15.02–61.22; p=0.27) and the difference between T4 and earlier stages (T2 and below) was non-significant (p=0.065). One might speculate that this may be due to the higher variability of gfDNA concentration seen for T4 (*Figure 2B*) that could be intrinsic to the T4 stage, as some tumors are more invasive than others, and if the tumor grows through the gastric wall and the serosa (*Fukuda et al., 2011*), it might perhaps be shedding DNA into adjacent peritoneal structures instead of just the gastric cavity, leading to a reduction of tumor-derived DNA in the gastric cavity. This contrasts with T0-T3 tumors that appear to continuously and preferentially release DNA inside the gastric cavity. In agreement with this possibility, we found that advanced tumors have increased gfDNA concentration as compared to early stages, a finding that might suggest that the origin of the gfDNA could perhaps derive from the tumor, but also from the tumor environment, from immune cell infiltrates, and the microbiota. However, our quantitative analysis of bacterial and human DNA in the gfDNA for a set of samples (n=180), including gastric cancer samples (n=64), showed no clear trends towards an increased bacterial DNA content, rendering that possibility unlikely (*Supplementary file 1*).

We also observed an intriguing correlation between gfDNA concentration and survival, with gastric cancer patients presenting lower gfDNA concentrations (set at ≤1.28 ng/µL) associated with a worse prognosis. We envisaged that this observation could be associated with poorer immune responses (immunologically 'cold tumors'), which was supported by the finding that patients with higher gfDNA concentrations (set at >1.28 ng/µL) more often presented with moderate-to-intense inflammatory infiltrates, whereas individuals with lower gfDNA usually presented minimal-to-moderate inflammatory infiltrates (p=0.001). Notably, two-thirds (16 out of 24) of the cases with increased gfDNA had

moderate/intense/elevated immune cell infiltrates, and all cases within this classification of inflammatory infiltrates were in the group of increased gfDNA (*Supplementary file 3*). This suggests that the elevated gfDNA concentration observed in subjects with better outcomes may originate from two primary sources: an active population of infiltrating immune cells, together with the tumor cells eliminated by these immune infiltrates.The results presented here suggest wide implications for the translational application of liquid biopsies with gfDNA concentration analysis in the diagnosis, staging, and management of gastric cancer patients. The most obvious barrier to the translational application of liquid biopsies and gfDNA measurement in the routine clinical setting is the need for an EGD to access it. However, the use of gfDNA may hold the highest benefit by increasing the value of information obtained from the initial EGD, which is – as mentioned previously – a procedure that all patients with gastric cancer will eventually undergo multiple times during their disease workup. Indeed, the suction of gastric fluids during endoscopy is readily performed during EGD, when the gastric walls must be free of fluids for a better visualization of the mucosa. Thus, the convenient incorporation of this liquid biopsy approach during EGD is cost-effective and a minor additional step to the procedure. The benefit is that the continuous proximity of gastric fluids to the tumor site portends a much-reduced dilution of gfDNA, some of it being tumor-derived DNA, which allows a better representation of tumor-related DNA alterations. In a comparative sense, liquid biopsy of gastric fluid with gfDNA measurement is similar to obtaining DNA from peritoneal lavage of patients with endometrial cancer, but indeed much harder to obtain, in which peritoneal fluid has higher mutant allelic fractions as compared to plasma (*Mayo-de-Las-Casas et al., 2020*), similar to previous findings in gastric cancer in our early work (*Pizzi et al., 2019*).

The results presented here suggest wide implications for the translational application of liquid biopsies with gfDNA concentration analysis in the diagnosis, staging, and management of gastric cancer patients. The most obvious barrier to the translational application of liquid biopsies and gfDNA measurement in the routine clinical setting is the need for an EGD to access it. However, the use of gfDNA may hold the highest benefit by increasing the value of information obtained from the initial EGD, which is – as mentioned previously – a procedure that all patients with gastric cancer will eventually undergo multiple times during their disease workup. Indeed, the suction of gastric fluids during endoscopy is readily performed during EGD, when the gastric walls must be free of fluids for a better visualization of the mucosa. Thus, the convenient incorporation of this liquid biopsy approach during EGD is cost-effective and a minor additional step to the procedure. The benefit is that the continuous proximity of gastric fluids to the tumor site portends a much-reduced dilution of gfDNA, some of it being tumor-derived DNA, which allows a better representation of tumor-related DNA alterations. In a comparative sense, liquid biopsy of gastric fluid with gfDNA measurement is similar to obtaining DNA from peritoneal lavage of patients with endometrial cancer, but indeed much harder to obtain, in which peritoneal fluid has higher mutant allelic fractions as compared to plasma (*Mayo-de-Las-Casas et al., 2020*), similar to previous findings in gastric cancer in our early work (*Pizzi et al., 2019*).

Much interest has been given to developing new tools for detecting diagnostic, prognostic, and predictive markers in gastric cancer (*Mayo-de-Las-Casas et al., 2020*). The area under the receiver operating characteristic (AUROC) analysis for liquid biopsy of gastric fluid with gfDNA analysis yielded a discrimination capability of 0.66, which is comparable to serological tumor biomarkers such as CEA (0.68), CA72-4 (0.67), or CA19-9 (0.64) that are currently used in standard oncological practice (*Tahara and Arisawa, 2015*; *Yu et al., 2016*; *Lin et al., 2020*). Moreover, the monitoring of gfDNA concentration may also be valuable for a closer follow-up and better diagnostic accuracy of patients with potentially premalignant diseases and/or at higher genetic/familial risk of developing gastric cancer. One might also speculate that our findings could perhaps also apply to other upper gastrointestinal malignant tumors such as esophageal cancer and/or gastroesophageal junction neoplasia, and future studies should rigorously evaluate those open possibilities.

This study represents an initial evaluation of the prognostic impact of gfDNA concentration in human subjects diagnosed with gastric cancer, as compared to other conditions such as pre-malignant lesions and peptic diseases. The participants were referred to diagnostic EGD, and their distribution in the diagnostic groups reflects the real-life case numbers seen in a single institution, a dedicated cancer center. Before receiving sedation for the EGD, participants were invited to join the study. Therefore, samples were collected prospectively, before diagnosis, from all subjects and no attempts to balance the groups were made to avoid introducing any potential selection bias. Limitations also

include the primer sets that allow a partial quantification of some bacterial groups (i.e. microbiome analysis), the limited number of pathology slides available from gastric cancer patients to investigate immune cell infiltrates (n=32), and the lack of a more comprehensive approach that would also allow a cell-type-specific deconvolution and a better explanation of the functional mechanisms that lead to increased gfDNA concentrations. The diagnostic potential of elevated gfDNA levels was not explored together with other known biomarkers such as CEA or CA19-9. Our data suggest that gfDNA is not only derived from tumor cells but may have expressive amounts of DNA derived from infiltrating immune cells. The precise determination of the presumed cell-of-origin of this increased gfDNA remains to be unequivocally demonstrated as it would have to rely on methods not used in this study, such as RNA-Seq and DNA-seq.

Moreover, the large patient cohort analyzed in this study was derived from a single specialized institution (a tertiary cancer center), and it is therefore unclear whether our findings might be broadly generalized; this open question should be addressed in further confirmatory studies from other hospital types and geographical regions.

Finally, for considering subjects with an established gastric cancer diagnosis, gfDNA has the potential to serve as a rapid, low-cost surrogate marker of the tumor immune microenvironment, whose characterization may have implications for predicting chemotherapy and immunotherapy response (*Jiang et al., 2018*; *Jiang et al., 2019*; *Chen et al., 2022*; *Espinosa-Carrasco et al., 2024*), and prognostic repercussions that, in the future, could be useful to treatment planning and risk-adapt therapeutic strategies. As we have gathered no data on gfDNA changes over time, conclusions on its utility for monitoring treatment response and/or predicting recurrence are currently limited but warrant further investigation.

Given that gastric cancer has currently no suitable tumor prognostic biomarker, one hopes that the initial results of this report will encourage the liquid biopsy of EGD-collected gastric fluids to evaluate gfDNA concentrations towards clinic-ready translational applications. Further gastric fluid studies may help revolutionize gastric cancer care, especially in places where specialties may not be readily available. Also, it may assist as a first-line exam that can be done to help detect cancer cases, with prognostic value. Although more research is needed, early data indicates this may be a way to provide access and availability for primary and recurrent gastric cancer.

# Materials and methods

**Key resources table**

| Reagent type (species) or resource | Designation | Source or reference | Identifiers | Additional information |
|---|---|---|---|---|
| Chemical compound, drug | Phenol Chloroform | Sigma Sigma | P4682 472476 | Used to remove proteins and lipids during DNA extraction. |
| Software, algorithm | R statistical environment GraphPad | R Core Team Prism | RRID:SCR_025679 RRID:SCR_002798 | Version 4.2.0, 2022 Version 10.4.1 (532) Dec. 2024 |
| Other | Proteinase-K | Bioline | BIO37037 | Enzyme used for protein digestion during DNA extraction. |

## Patients

This single-center convenience study performed at the A. C. Camargo Cancer Center, São Paulo, Brazil, included all patients who provided informed consent and met the inclusion criteria of being at least 18 years old and having a clinical indication for diagnostic EGD. Samples were collected at diagnosis, during standard EGD, from February 2016 to March 2021. EGDs were only performed after referral from an independent physician (i.e. not related to the study) to assess abdominal symptoms, under standard sedation. Samples were obtained after either patients or their guardians signed a written informed consent form approved by the Clinical Research Committee, the Ethics Research Committee, and the Institutional Review Board (IRB; protocol #2134/15). Diagnostic and epidemiological data were collected as described (*Bartelli et al., 2019*). Samples and analyses were excluded if requested by the participants (even after the signature of a written informed consent), or if other unrelated diagnoses were found (e.g. previous gastrectomy or presence of other non-gastric cancer).

## Gastric fluid collection, DNA extraction, and quantification

As usual and customary for upper endoscopic examination, all subjects were fasting for 8–12 hr. At the beginning of the procedures, gastric fluids that are routinely removed to allow better examination of the mucosa were collected in sterile plastic containers attached to the endoscope suction channel and kept on ice until pH measurement and neutralization. Samples were divided into aliquots and kept frozen at –20°C (each aliquot was thawed just once), until DNA was extracted from an equal volume of gastric fluid (800 µl) by proteinase-K digestion, followed by phenol-chloroform extraction and ethanol DNA precipitation. DNA was subsequently resuspended in 100 µl of sterile nuclease-free double-deionized water and quantified by Qubit fluorometry (Thermo Fisher). DNA concentrations found for each subject, as well as clinical/pathological data and patient demographics related to this article are freely available at: https://doi.org/10.5061/dryad.bzkh189qz.

## Quantification of human- versus bacteria-derived DNA in gastric fluids

For a representative fraction of the liquid biopsies (n=180, 19.1%), including all 10 control subjects with no clinical findings (i.e. no pathologies detected on EGD) along with peptic diseases (n=51), premalignant lesions (n=55), and gastric cancer cases (n=64), we evaluated the proportion of human- versus bacteria-derived DNA in gastric fluids by using quantitative PCR (*de Albuquerque et al., 2022*).

## Pathological analysis of inflammatory cell infiltrates in tissue biopsies

Tissue biopsies were available from a subset of liquid biopsies evaluated for the gfDNA content, allowing the investigation of inflammatory cell infiltrates. In such cases, pathological tissue analyses were done independently by two expert cancer pathologists, both specialized in upper gastrointestinal tract neoplasia, who were blinded to the gfDNA concentration results from the liquid biopsies. Discrepant cases for the immune infiltration levels (n=3) were revised and discussed, and a consensus was achieved. After the microscopic identification of tumor cells and the histological classification of gastric cancer, the presence and intensity of inflammatory cell infiltrates in the neoplastic lesions were also evaluated, as well as their patterns in the tumor stroma (e.g. diffuse, microscopic lymphoid aggregates, lymphoid aggregates with the formation of hyperplastic germinal centers). We have set the predominant cellular composition of the inflammatory infiltrates and quantified the infiltrate/tumor stroma ratios as follows: (i) minimal/rare – rare inflammatory cells in the tumor stroma; (ii) moderate – intermediate cellularity, no distortion of the epithelial component and without formation of luminal micro-abscesses, and (iii) intense/elevated – large amounts of inflammatory cells, making it difficult to visualize the stroma, with the formation of luminal micro-abscesses and/or epithelial thinning. We have also evaluated the presence of lymphoid aggregates, necrosis/ulcer, and neutrophilic infiltrates. Areas directly related to ulcer formation and/or granulation tissue were not considered in the polymorphonuclear component assessment.

## Statistical analysis

No formal sample size calculation was performed as in this exploratory study we used convenience sampling, recruiting all consecutive eligible patients who met the inclusion criteria and consented to participate during the study period. Baseline gfDNA concentration values for patient groups are presented as mean ± standard deviation or as median and interquartile range, as appropriate. Analyses were performed in the R statistical environment (version 4.2.0) as described by the *R Development Core Team, 2022*. Data distribution was assessed using the Shapiro–Wilk test and visual inspection of Q–Q plots. Normally distributed data were analyzed using one-way ANOVA with Tukey's HSD post hoc test, whereas non-normally distributed data were analyzed using the Kruskal–Wallis test with Dunn's post hoc test. Parametric tests were applied to normally distributed data, and non-parametric alternatives were used otherwise. Optimal cutoffs for gfDNA concentrations to stratify survival groups were determined using maximally selected rank statistics (maxstat.test from the R package maxstat; *Hothorn, 2017*). Distributions of gfDNA concentrations among the four patient groups (normal, peptic ulcer disease, preneoplastic disease, and gastric cancer) were compared using the Kruskal–Wallis test, followed by pairwise Mann–Whitney U tests with Benjamini–Hochberg adjustment of p-values. ROC curves and the corresponding AUC were obtained using logistic regression analysis. Results were considered statistically significant when $p \leq 0.05$.

## Acknowledgements

Authors acknowledge the support given by the institutional Tumor Bank from AC Camargo Cancer Center. We thank Dr. Christian Abnet and Dr. M Constanza Camargo (NCI, NIH, USA) for their critical review of this manuscript.

## Additional information

### Competing interests

Wadih Arap: Reviewing editor, eLife. The other authors declare that no competing interests exist.

### Funding

| Funder | Grant reference number | Author |
| --- | --- | --- |
| Fundação de Amparo à Pesquisa do Estado de São Paulo | 2014/26897-0 | Emmanuel Dias-Neto |
| Ministério da Saúde | PRONON/DECIT | Emmanuel Dias-Neto |
| Levy-Longenbaugh | Donor-Advised Fund | Renata Pasqualini Wadih Arap |
| Fundação de Amparo à Pesquisa do Estado de São Paulo | 2018/14267-2 | Emmanuel Dias-Neto |
| Fundação de Amparo à Pesquisa do Estado de São Paulo | 2018/02972-3 | Emmanuel Dias-Neto |
| Ministério da Saúde | SIPAR 2500.035-167/2015-23 | Emmanuel Dias-Neto |

The funders had no role in study design, data collection and interpretation, or the decision to submit the work for publication.

### Author contributions

Francine C Cadoná, Data curation, Formal analysis, Investigation, Visualization, Writing – original draft, Writing – review and editing; Thais F Bartelli, Formal analysis, Investigation, Writing – review and editing; Adriane G Pelosof, Investigation, Methodology, Writing – original draft, Writing – review and editing, Sample collection; Claudia Z Sztokfisz, Formal analysis, Investigation, Writing – review and editing, Sample collection; Adriana P Bueno, Data curation, Investigation, Methodology, Writing – review and editing, Pathology analysis; Luana Batista do Carmo dos Santos, Formal analysis, Investigation, Sample and clinical data collection; Gabriela P Branco, Investigation, Methodology, Sample and clinical data collection; Gabriel Oliveira dos Santos, Investigation, Writing – review and editing, pathology analysis; Warley A Nunes, Formal analysis, Investigation, Writing – review and editing, Pathology analysis; Fernanda A Pintor, Investigation, Writing – review and editing, Clinical data collection; Laís Lie Senda de Abrantes, Investigation, Clinical data collection; Alexandre Defelicibus, Writing – review and editing, Statistical support; Luiz Gonzaga Vaz Coelho, Writing – review and editing, Discussion of findings and clinical relevance; Marcis Leja, Writing – review and editing, Discussion of findings and clinical relevance; Haejin In, Writing – review and editing, Discussion of findings and clinical relevance; Sharon Li, Writing – review and editing, Discussion of findings and clinical relevance; Howard Hochster, Writing – review and editing, Discussion of findings and clinical relevance; Felipe JF Coimbra, Investigation, Writing – review and editing, Discussion of findings and clinical relevance; Rodrigo Drummond, Writing – review and editing, Statistical support; Israel Tojal Da Silva, Data curation, Investigation, Writing – review and editing, Statistical support; Ravi J Chokshi, Writing – review and editing, Discussion of findings and clinical relevance; Renata Pasqualini, Writing – review and editing, Discussion of findings and clinical relevance; Wadih Arap, Supervision, Writing – review and editing, Discussion of findings and clinical relevance; Diana N Nunes, Conceptualization, Investigation, Writing – original draft, Project administration, Writing – review and editing, Discussion of

findings and clinical relevance; Emmanuel Dias-Neto, Conceptualization, Formal analysis, Supervision, Funding acquisition, Investigation, Methodology, Writing – original draft, Project administration, Writing – review and editing

### Author ORCIDs
Israel Tojal Da Silva ⓘ https://orcid.org/0000-0002-4687-1499
Emmanuel Dias-Neto ⓘ https://orcid.org/0000-0001-5670-8559

### Ethics
Samples were obtained after either patients or their guardians signed a written informed consent form approved by the Clinical Research Committee, the Ethics Research Committee, and the Institutional Review Board (IRB) of the A.C.Camargo cancer Center (protocol #2134/15).

Reviewer #1 (Public review): https://doi.org/10.7554/eLife.107103.3.sa1
Reviewer #2 (Public review): https://doi.org/10.7554/eLife.107103.3.sa2
Author response https://doi.org/10.7554/eLife.107103.3.sa3

---

## Additional files

### Supplementary files
MDAR checklist

Supplementary file 1. Analysis of CT values for bacteria and human target genes.

Supplementary file 2. Performance metrics of the ROC/AUC curves shown in *Figure 3*.

Supplementary file 3. gfDNA concentration and intensity of inflammatory cell infiltrates.

### Data availability
The datasets generated and/or analyzed during the current study are available either in this published article or in the following publicly accessible repository: https://doi.org/10.5061/dryad.bzkh189qz.

The following dataset was generated:

| Author(s) | Year | Dataset title | Dataset URL | Database and Identifier |
|---|---|---|---|---|
| Cadoná FC, Bartelli TF, Pelosof AG, Sztokfisz CZ, Bueno AP, Santos LB, Branco GP, Santos GO, Nunes WA, Pintor FA, Abrantes LL, Defelicibus A, Coelho LV, Leja M, In H, Li S, Hochster H, Coimbra FJ, Drummond R, Silva I, Chokshi R, Pasqualini R, Arap W, Nunes DN, Dias-Neto E | 2025 | Patient data, demographics, clinical data and DNA concentration | https://doi.org/10.5061/dryad.bzkh189qz | Dryad Digital Repository, 10.5061/dryad.bzkh189qz |

---

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
