## [Editor Report · eLife Assessment]

This **valuable** work substantially advances our understanding of prognostic value of total gfDNA in gastric cancer. The evidence supporting the conclusions is **solid**, supported by a large, well-classified patient cohort and controlled clinical variables. The work will be of broad interest to scientists and clinical pathologist working in the field of gastric cancer.

---

## [Referee Report · Reviewer #1 (Public review)]

The study analyzes the gastric fluid DNA content identified as a potential biomarker for human gastric cancer. However, the study lacks overall logicality, and several key issues require improvement and clarification. In the opinion of this reviewer, some major revisions are needed:

(1) This manuscript lacks a comparison of gastric cancer patients' stages with PN and N+PD patients, especially T0-T2 patients.

(2) The comparison between gastric cancer stages seems only to reveal the difference between T3 patients and early-stage gastric cancer patients, which raises doubts about the authenticity of the previous differences between gastric cancer patients and normal patients, whether it is only due to the higher number of T3 patients.

(3) The prognosis evaluation is too simplistic, only considering staging factors, without taking into account other factors such as tumor pathology and the time from onset to tumor detection.

(4) The comparison between gfDNA and conventional pathological examination methods should be mentioned, reflecting advantages such as accuracy and patient comfort.

(5) There are many questions in the figures and tables. Please match the Title, Figure legends, Footnote, Alphabetic order, etc.

(6) The overall logicality of the manuscript is not rigorous enough, with few discussion factors, and cannot represent the conclusions drawn.

Comments on revisions:

The authors have addressed all concerns in the revision.

---

## [Referee Report · Reviewer #2 (Public review)]

Summary

The authors aimed to evaluate whether total DNA concentration in gastric fluid (gfDNA) collected during routine endoscopy could serve as a diagnostic and prognostic biomarker for gastric cancer. Using a large cohort (n=941), they reported elevated gfDNA in gastric cancer patients, an unexpected association with improved survival, and a positive correlation with immune cell infiltration.

Strengths

The study benefits from a substantial sample size, clear patient stratification, and control of key clinical confounders. The method is simple and clinically feasible, with preliminary evidence linking gfDNA to immune infiltration.

Weaknesses

(1) While the study identifies gfDNA as a potential prognostic tool, the evidence remains preliminary. Unexplained survival associations and methodological gaps weaken support for the conclusions.

(2) The paradoxical association between high gfDNA and better survival lacks mechanistic validation. The authors acknowledge but do not experimentally distinguish tumor vs. immune-derived DNA, leaving the biological basis speculative.

(3) Pre-analytical variables were noted but not systematically analyzed for their impact on gfDNA stability.

Comments on revisions:

To enhance the completeness and credibility of this research, it is essential to clarify the biological origin of gastric fluid DNA and validate these preliminary findings through a prospective, longitudinal study design.

---

## [Author Response]

The following is the authors’ response to the original reviews.

**Reviewer #1 (Public review):**
“The study analyzes the gastric fluid DNA content identified as a potential biomarker for human gastric cancer. However, the study lacks overall logicality, and several key issues require improvement and clarification. In the opinion of this reviewer, some major revisions are needed:”(1) “This manuscript lacks a comparison of gastric cancer patients' stages with PN and N+PD patients, especially T0-T2 patients.”

We are grateful for this astute remark. A comparison of gfDNA concentration among the diagnostic groups indicates a trend of increasing values as the diagnosis progresses toward malignancy. The observed values for the diagnostic groups are as follows:

**Author response table 1. sa3table1:** 

Diagnosis group	Mean gfDNA and 95% Confidence Interval
Normal mucosa and peptic diseases (N=606)	10.77ng//muL; 95% CI: 9.23 to 12.33;
Preneoplastic conditions (N=99)	10.10ng//muL; 95% Cl: 7.59 to 12.60
T2 and below (N=65)	15.12ng//muL; 95% Cl: 9.73 to 20.50
T3 and above (N=154)	30.75ng//muL; 95% Cl: 14.24 to 36.62

The chart below presents the statistical analyses of the same diagnostic/tumor-stage groups (One-Way ANOVA followed by Tukey’s multiple comparison tests). It shows that gastric fluid gfDNA concentrations gradually increase with malignant progression. We observed that the initial tumor stages (T0 to T2) exhibit intermediate gfDNA levels, which in this group is significantly lower than in advanced disease (p = 0.0036), but not statistically different from non-neoplastic disease (p = 0.74).

(2) “The comparison between gastric cancer stages seems only to reveal the difference between T3 patients and early-stage gastric cancer patients, which raises doubts about the authenticity of the previous differences between gastric cancer patients and normal patients, whether it is only due to the higher number of T3 patients.”

We appreciate the attention to detail regarding the numbers analyzed in the manuscript. Importantly, the results are meaningful because the number of subjects in each group is comparable (T0-T2, N = 65; T3, N = 91; T4, N = 63). The mean gastric fluid gfDNA values (ng/µL) increase with disease stage (T0-T2: 15.12; T3-T4: 30.75), and both are higher than the mean gfDNA values observed in non-neoplastic disease (10.81 ng/µL for N+PD and 10.10 ng/µL for PN). These subject numbers in each diagnostic group accurately reflect real-world data from a tertiary cancer center.

(3) “The prognosis evaluation is too simplistic, only considering staging factors, without taking into account other factors such as tumor pathology and the time from onset to tumor detection.”

Histopathological analyses were performed throughout the study not only for the initial diagnosis of tissue biopsies, but also for the classification of Lauren’s subtypes, tumor staging, and the assessment of the presence and extent of immune cell infiltrates. Regarding the time of disease onset, this variable is inherently unknown--by definition--at the time of a diagnostic EGD. While the prognosis definition is indeed straightforward, we believe that a simple, cost-effective, and practical approach is advantageous for patients across diverse clinical settings and is more likely to be effectively integrated into routine EGD practice.

(4) “The comparison between gfDNA and conventional pathological examination methods should be mentioned, reflecting advantages such as accuracy and patient comfort. “

We wish to reinforce that EGD, along with conventional histopathology, remains the gold standard for gastric cancer evaluation. EGD under sedation is routinely performed for diagnosis, and the collection of gastric fluids for gfDNA evaluation does not affect patient comfort. Thus, while gfDNA analysis was evidently not intended as a diagnostic EGD and biopsy replacement, it may provide added prognostic value to this exam.

(5) “There are many questions in the figures and tables. Please match the Title, Figure legends, Footnote, Alphabetic order, etc. “

We are grateful for these comments and apologize for the clerical oversight. All figures, tables, titles and figure legends have now been double-checked.

(6) “The overall logicality of the manuscript is not rigorous enough, with few discussion factors, and cannot represent the conclusions drawn. “

We assume that the unusual wording remark regarding “overall logicality” pertains to the rationale and/or reasoning of this investigational study. Our working hypothesis was that during neoplastic disease progression, tumor cells continuously proliferate and, depending on various factors, attract immune cell infiltrates. Consequently, both tumor cells and immune cells (as well as tumor-derived DNA) are released into the fluids surrounding the tumor at its various locations, including blood, urine, saliva, gastric fluids, and others. Thus, increases in DNA levels within some of these fluids have been documented and are clinically meaningful. The concurrent observation of elevated gastric fluid gfDNA levels and immune cell infiltration supports the hypothesis that increased gfDNA—which may originate not only from tumor cells but also from immune cells—could be associated with better prognosis, as suggested by this study of a large real-world patient cohort.

In summary, we thank Reviewer #1 for his time and effort in a constructive critique of our work.

**Reviewer #2 (Public review):**
Summary:“The authors investigated whether the total DNA concentration in gastric fluid (gfDNA), collected via routine esophagogastroduodenoscopy (EGD), could serve as a diagnostic and prognostic biomarker for gastric cancer. In a large patient cohort (initial n=1,056; analyzed n=941), they found that gfDNA levels were significantly higher in gastric cancer patients compared to non-cancer, gastritis, and precancerous lesion groups. Unexpectedly, higher gfDNA concentrations were also significantly associated with better survival prognosis and positively correlated with immune cell infiltration. The authors proposed that gfDNA may reflect both tumor burden and immune activity, potentially serving as a cost-effective and convenient liquid biopsy tool to assist in gastric cancer diagnosis, staging, and follow-up.”Strengths:“This study is supported by a robust sample size (n=941) with clear patient classification, enabling reliable statistical analysis. It employs a simple, low-threshold method for measuring total gfDNA, making it suitable for large-scale clinical use. Clinical confounders, including age, sex, BMI, gastric fluid pH, and PPI use, were systematically controlled. The findings demonstrate both diagnostic and prognostic value of gfDNA, as its concentration can help distinguish gastric cancer patients and correlates with tumor progression and survival. Additionally, preliminary mechanistic data reveal a significant association between elevated gfDNA levels and increased immune cell infiltration in tumors (p=0.001).”

Reviewer #2 has conceptually grasped the overall rationale of the study quite well, and we are grateful for their assessment and comprehensive summary of our findings.

Weaknesses:(1) “The study has several notable weaknesses. The association between high gfDNA levels and better survival contradicts conventional expectations and raises concerns about the biological interpretation of the findings.“

We agree that this would be the case if the gfDNA was derived solely from tumor cells. However, the findings presented here suggest that a fraction of this DNA would be indeed derived from infiltrating immune cells. The precise determination of the origin of this increased gfDNA remains to be achieved in future follow-up studies, and these are planned to be evaluated soon, by applying DNA- and RNA-sequencing methodologies and deconvolution analyses.

(2) “The diagnostic performance of gfDNA alone was only moderate, and the study did not explore potential improvements through combination with established biomarkers. Methodological limitations include a lack of control for pre-analytical variables, the absence of longitudinal data, and imbalanced group sizes, which may affect the robustness and generalizability of the results.“

Reviewer #2 is correct that this investigational study was not designed to assess the diagnostic potential of gfDNA. Instead, its primary contribution is to provide useful prognostic information. In this regard, we have not yet explored combining gfDNA with other clinically well-established diagnostic biomarkers. We do acknowledge this current limitation as a logical follow-up that must be investigated in the near future.

Moreover, we collected a substantial number of pre-analytical variables within the limitations of a study involving over 1,000 subjects. Longitudinal samples and data were not analyzed here, as our aim was to evaluate prognostic value at diagnosis. Although the groups are imbalanced, this accurately reflects the real-world population of a large endoscopy center within a dedicated cancer facility. Subjects were invited to participate and enter the study before sedation for the diagnostic EGD procedure; thus, samples were collected prospectively from all consenting individuals.

Finally, to maintain a large, unbiased cohort, we did not attempt to balance the groups, allowing analysis of samples and data from all patients with compatible diagnoses (please see Results: Patient groups and diagnoses).

(3) “Additionally, key methodological details were insufficiently reported, and the ROC analysis lacked comprehensive performance metrics, limiting the study's clinical applicability.“

We are grateful for this useful suggestion. In the current version, each ROC curve (Supplementary Figures 1A and 1B) now includes the top 10 gfDNA thresholds, along with their corresponding sensitivity and specificity values (please see Suppl. Table 1). The thresholds are ordered from-best-to-worst based on the classic Youden’s J statistic, as follows:

Youden Index = specificity + sensitivity – 1 [Youden WJ. Index for rating diagnostic tests. *Cancer* 3:32-35, 1950. PMID: 15405679]. We have made an effort to provide all the key methodological details requested, but we would be glad to add further information upon specific request.

**Reviewer #1 (Recommendations for the authors):**
The authors should pay attention to ensuring uniformity in the format of all cited references, such as the number of authors for each reference, the journal names, publication years, volume numbers, and page number formats, to the best extent possible.

Thank you for pointing this inconsistency. All cited references have now been revisited and adjusted properly. We apologize for this clerical oversight.

**Reviewer #2 (Recommendations for the authors):**
(1) “High gfDNA levels were surprisingly linked to better survival, which conflicts with the conventional understanding of cfDNA as a tumor burden marker. Was any qualitative analysis performed to distinguish DNA derived from immune cells versus tumor cells?“

Tumor-derived DNA is certainly present in gfDNA, as our group has unequivocally demonstrated in a previous publication [Pizzi M. P., et al. (2019) Identification of DNA mutations in gastric washes from gastric adenocarcinoma patients: Possible implications for liquid biopsies and patient follow-up *Int J Cancer* 145:1090–1097. DOI: 10.1002/ijc.32114]. However, in the present manuscript, our data suggest that gfDNA may also contain DNA derived from infiltrating immune cells. This may also be the case for other malignancies, and qualitative deconvolution studies could provide more informative information. To achieve this, DNA sequencing and RNA-Seq analyses may offer relevant evidence. Our study should be viewed as an original and preliminary analysis that may encourage such quantitative and qualitative studies in biofluids from cancer patients. Currently, this is a simple approach (which might be its essential beauty), but we hope to investigate this aspect further in future studies.

(2) “The ROC curve AUC was 0.66, indicating only moderate discrimination ability. Did the authors consider combining gfDNA with markers such as CEA or CA19-9 to improve diagnostic accuracy?“

This is indeed a logical idea, which shall certainly be explored in planned follow-up studies.

(3) “DNA concentration could be influenced by non-biological factors, including gastric fluid pH, sampling location, time delay, or freeze-thaw cycles. Were these operational variables assessed for their effect on data stability?“

We appreciate the rigor of the evaluation. Yes, information regarding gastric fluid pH was collected. All samples were collected from the stomach during EGD procedure. Samples were divided in aliquots and were thawed only once. This information is now provided in the updated manuscript text.

(4) “This cross-sectional study lacks data on gfDNA changes over time, limiting conclusions on its utility for monitoring treatment response or predicting recurrence.“

Again, temporal evaluation is another excellent point, and it will be the subject of future analyses. In this exploratory study, samples were collected at diagnosis, at a single point. We have not obtained serial samples, as participants received appropriate therapy soon following diagnosis.

(5) The normal endoscopy group included only 10 patients, the precancerous lesion group 99 patients, while the gastritis group had 596 patients. Such uneven sample sizes may affect statistical reliability and generalizability. Has weighted analysis or optimized sampling been considered for future studies?“

Yes, in future studies this analysis will be considered, probably by employing stratified random sampling with relevant patient attributes recorded.

(6) “The SciScore was only 2 points, indicating that key methodological details such as inclusion/exclusion criteria, randomization, sex variables, and power calculation were not clearly described. It is recommended that these basic research elements be supplemented in the Methods section. “

This was an exploratory research, the first of its kind, to evaluate prognostic potential of gfDNA in the context of gastric cancer. Patients were not included if they did not sign the informed consent or excluded if they withdrew after consenting. Other exclusion criteria included diagnoses of conditions such as previous gastrectomy or esophagectomy, or the presence of non-gastric malignancies. Randomization and power analyses were not applicable, as no prior data were available regarding gfDNA concentration values or its diagnostic/prognostic potential. All subjects, regardless of sex, were invited to participate without discrimination or selection.

(7) “Although a ROC curve was provided in the supplementary materials (Supplementary Figure 1), only the curve and AUC value were shown without sensitivity, specificity, predictive values, or cutoff thresholds. The authors are advised to provide a full ROC performance assessment to strengthen the study's clinical relevance.

These data are now given alongside the ROC curves in the Supplementary Information section, specifically in Supplementary Figure 1 and in the newly added Supplementary Table 1.

We thank Reviewer #2 for an insightful and positive overall assessment of our work.